# Magnesium sulphate reduces tertiary gliosis but does not improve EEG recovery or white or grey matter cell survival after asphyxia in preterm fetal sheep

Robert Galinsky[1,2,3] ⬤, Simerdeep K. Dhillon[1], Sharmony B. Kelly[2,3], Guido Wassink[1], Joanne O. Davidson[1] ⬤, Christopher A. Lear[1] ⬤, Lotte G. van den Heuij[1], Laura Bennet[1] ⬤ and Alistair J. Gunn[1] ⬤

[1] *Department of Physiology, University of Auckland, Auckland, New Zealand*
[2] *The Ritchie Centre, Hudson Institute of Medical Research, Clayton, Victoria, Australia*
[3] *Department of Obstetrics and Gynaecology, Monash University, Victoria, Australia*

Handling Editors: Harold Schultz & Janna Morrison

The peer review history is available in the Supporting Information section of this article (https://doi.org/10.1113/JP284381#support-information-section).

- Reduced numbers of total Olig-2 OLs
- Did not improve survival of mature myelinating CC1 OLs

**Intragyral and periventricular oligodendrocytes (OLs)**

**Neurons**

Did not improve neuronal survival in the premotor cortex and striatum

**Myelin density**
Intermediate improvement in myelin density

$MgSO_4$ + hypoxia ischaemia

**Astrocytes**

**Microglia**

↓ astrocytosis and microgliosis in the premotor cortex, striatum, and periventricular and intragyral white matter tracts

The Journal of **Physiology**

**Abstract**   Maternal magnesium sulphate ($MgSO_4$) treatment is widely recommended before preterm birth for neuroprotection. However, this is controversial because there is limited evidence

**Robert Galinsky** is head of the Perinatal Inflammation and Neurophysiology Group, which is co-located at the Ritchie Centre, Hudson Institute of Medical Research and the Department of Obstetrics and Gynaecology, Monash University in Melbourne, Victoria, Australia. Rob and his team focus on understanding the cellular and physiological mechanisms that underpin perinatal brain injury. Using this information, the Perinatal Inflammation and Neurophysiology Group are discovering new and improved drug targets for the treatment of perinatal brain injury and developing physiological biomarkers and brain-imaging techniques to improve detection of injury. They use a multidisciplinary approach that includes fundamental neuroscience, physiology, medical imaging and bioengineering. Rob undertook his postdoctoral training at Auckland University with the Fetal Physiology and Neuroscience Group, co-led by Professors Laura Bennet and Alistair Gunn.

The Journal of Physiology

that $MgSO_4$ provides long-term neuroprotection. Preterm fetal sheep (104 days gestation; term is 147 days) were assigned randomly to receive sham occlusion with saline infusion ($n = 6$) or i.v. infusion with $MgSO_4$ ($n = 7$) or vehicle (saline, $n = 6$) from 24 h before hypoxia–ischaemia induced by umbilical cord occlusion until 24 h after occlusion. Sheep were killed after 21 days of recovery, for fetal brain histology. Functionally, $MgSO_4$ did not improve long-term EEG recovery. Histologically, in the premotor cortex and striatum, $MgSO_4$ infusion attenuated post-occlusion astrocytosis ($GFAP^+$) and microgliosis but did not affect numbers of amoeboid microglia or improve neuronal survival. In the periventricular and intragyral white matter, $MgSO_4$ was associated with fewer total ($Olig\text{-}2^+$) oligodendrocytes compared with vehicle + occlusion. Numbers of mature ($CC1^+$) oligodendrocytes were reduced to a similar extent in both occlusion groups compared with sham occlusion. In contrast, $MgSO_4$ was associated with an intermediate improvement in myelin density in the intragyral and periventricular white matter tracts. In conclusion, a clinically comparable dose of $MgSO_4$ was associated with moderate improvements in white and grey matter gliosis and myelin density but did not improve EEG maturation or neuronal or oligodendrocyte survival.

(Received 15 January 2023; accepted after revision 9 March 2023; first published online 30 March 2023)

**Corresponding author** R. Galinsky: The Ritchie Centre, Hudson Institute of Medical Research, 27–31 Wright Street, Clayton, VIC 3168, Australia. Email: robert.galinsky@hudson.org.au

**Abstract figure legend** Schematic diagram summarizing key study outcomes. In preterm fetal sheep exposed to hypoxia–ischaemia (HI), a clinically comparable dose of $MgSO_4$ was associated with attenuated astrocytosis and microgliosis in the premotor cortex and striatum but did not improve neuronal survival after recovery to term-equivalent age, 21 days after HI. Magnesium sulphate was associated with exacerbated loss of oligodendrocytes in the periventricular and intragyral white matter tracts, whereas mature, myelinating oligodendrocytes were reduced to a similar extent in both HI groups. In the same regions, $MgSO_4$ was associated with an intermediate improvement in myelin density.

## Key points

- Magnesium sulphate is widely recommended before preterm birth for neuroprotection; however, there is limited evidence that magnesium sulphate provides long-term neuroprotection.
- In preterm fetal sheep exposed to hypoxia–ischaemia (HI), $MgSO_4$ was associated with attenuated astrocytosis and microgliosis in the premotor cortex and striatum but did not improve neuronal survival after recovery to term-equivalent age, 21 days after HI.
- Magnesium sulphate was associated with loss of total oligodendrocytes in the periventricular and intragyral white matter tracts, whereas mature, myelinating oligodendrocytes were reduced to a similar extent in both occlusion groups. In the same regions, $MgSO_4$ was associated with an intermediate improvement in myelin density.
- Functionally, $MgSO_4$ did not improve long-term recovery of EEG power, frequency or sleep stage cycling.
- A clinically comparable dose of $MgSO_4$ was associated with moderate improvements in white and grey matter gliosis and myelin density but did not improve EEG maturation or neuronal or oligodendrocyte survival.

## Introduction

Magnesium sulphate ($MgSO_4$) is now widely recommended for neuroprotection for preterm birth in many countries (Committee on Obstetric Practice & the Society for Maternal-Fetal Medicine, 2010; Magee et al., 2011). This recommendation is based on meta-analysis of randomized controlled trials of antenatal administration of $MgSO_4$ to women at risk of preterm birth that found that this intervention is associated with a small but significant reduction in the risk of cerebral palsy [relative risk (RR) 0.61; 95% confidence interval (CI) 0.44–0.92] and motor dysfunction (RR 0.61; 95% CI 0.44–0.85) in early childhood (Doyle et al., 2009). However, there was no significant effect on overall death and disability

(RR 0.94; 95% CI 0.78–1.12), raising the possibility that exposure to $MgSO_4$ might have been associated with a partial shift between outcomes rather than overall improvement.

Preterm brain injury is multifactorial (Galinsky, Davidson et al., 2018; Galinsky, Lear et al., 2018). Asphyxia, as shown by metabolic acidosis and need for resuscitation at birth, remains common among preterm babies and is associated with increased risk of death, sub-cortical brain injury and disability (Barkovich & Sargent, 1995; Corchia et al., 2013; Kerstjens et al., 2012; Low et al., 2003; Randolph et al., 2014; Reid et al., 2014; Sukhov et al., 2012). Furthermore, excessive glutaminergic excitation and CNS inflammation during the evolution of hypoxic ischaemic encephalopathy (HIE) are widely implicated in preterm white and grey matter injury (Fraser et al., 2008; Oka et al., 1993; Volpe, 2009).

The most likely mechanism for neuroprotection with magnesium is through its physiological role as an end-ogenous inhibitor of NMDA receptor activation by excitatory amino acids, such as glutamate (Galinsky et al., 2019; Zeevalk & Nicklas, 1992). This is supported by evidence in preterm fetal sheep and neonatal piglets for suppression of baseline EEG activity and reduced seizures with magnesium infusions after experimentally induced HIE (Bennet et al., 2018; Galinsky et al., 2015; Galinsky, Dhillon et al., 2018; Lingam et al., 2019). There is some evidence for other mechanisms, such as reduced inflammation (Sugimoto et al., 2012), inhibition of free radical production (Maulik et al., 1999), neuro-nal cell membrane stabilization (Hoffman et al., 1994) and improved cardiovascular stability (Galinsky et al., 2015; Galinsky, Dhillon et al., 2018; Shokry et al., 2010).

Despite these potential beneficial effects, systematic reviews of the preclinical and clinical literature have found that the effects of $MgSO_4$ treatment during or after hypoxia–ischaemia or perinatal infection/inflammation on neural outcomes were inconsistent between studies (Galinsky et al., 2014; Galinsky, Dean et al., 2020). In part, the inconsistent effects in small animal studies probably reflected confounding with iatrogenic hypo-thermia (Galinsky et al., 2014; Galinsky, Dean et al., 2017, 2020). There was a modest to no histological benefit in large animal studies at term-equivalent age (de Haan et al., 1997; Galinsky et al., 2014). The only study in preterm-equivalent fetal sheep to date (Galinsky, Draghi et al., 2017) demonstrated that $MgSO_4$ did not reduce asphyxia-induced brain injury and, of concern, exacerbated loss of oligodendrocytes after 3 days of recovery. Critically, no large animal studies have examined whether preterm exposure to $MgSO_4$ improves myelination after recovery to term-equivalent brain maturation. Supporting the preclinical evidence, in clinical studies, follow-up to school age suggests no significant improvement in neurodevelopmental outcomes (Chollat et al., 2014; Doyle et al., 2014), although these studies are relatively small and therefore under-powered to detect small differences between treatment and placebo.

These data highlight the need to understand better whether antenatal $MgSO_4$ exposure might mitigate cerebral injury in the preterm brain after recovery to term-equivalent maturation. Thus, the aim of this study was to determine whether $MgSO_4$ can improve maturation of EEG activity and mitigate white and grey matter injury in preterm fetal sheep after 3 weeks of recovery from asphyxia at 0.7 of gestation. At this gestational age, neural maturation is broadly equivalent to 28−30 weeks of human development (Barlow, 1969).

## Methods

### Ethics approval

All procedures were approved by the Animal Ethics Committee of The University of Auckland under the New Zealand Animal Welfare Act and the Code of Ethical Conduct for animals in research established by the Ministry of Primary Industries, Government of New Zealand (AEC approval number 1942). The experiments are reported in accordance with the ARRIVE guidelines for reporting animal research (Percie du Sert et al., 2020). All procedures comply with the guidelines of *The Journal of Physiology* (Grundy, 2015).

### Fetal surgery

Nineteen Romney/Suffolk fetal sheep underwent aseptic surgery between 97 and 99 days of gestation (term = 147 days). Food but not water was withdrawn 18 h before surgery. Ewes were given long-acting oxytetracycline (20 mg/kg; Phoenix Pharm, Auckland, New Zealand) I.M. 30 min before the start of surgery. Anaesthesia was induced by I.V. injection of propofol (5 mg/kg; AstraZeneca, Auckland, New Zealand) and maintained using 2−3% isoflurane in $O_2$ (Bomac Animal Health, NSW, Australia). During surgery, ewes received an I.V. infusion of isotonic saline (250 ml/h) to maintain fluid balance. The depth of anaesthesia, maternal heart rate and respiration were continuously monitored by trained anaesthetic staff.

### Instrumentation

In brief, following a maternal mid-line abdominal incision, the fetus was exposed, and polyvinyl catheters were inserted in the right brachial artery and vein and amniotic cavity. A pair of electrodes was sewn over the fetal chest to measure the fetal ECG. An inflatable silicone rubber occluder (*In Vivo* Metric, Healdsburg, CA, USA)

was placed loosely around the umbilical cord. Two pairs of EEG electrodes (AS633-7SSF; Cooner Wire, Chatsworth, CA, USA) were placed through burr holes onto the dura over the parasagittal parietal cortices (5 and 10 mm anterior to bregma and 5 mm lateral) and secured with cyanoacrylate glue. A reference electrode was sewn over the occiput. All fetal leads were exteriorized through the maternal flank. Antibiotics (gentamicin; 80 mg; Rousell, Auckland, New Zealand) were administered into the amniotic sac before closure of the uterus. A maternal long saphenous vein was catheterized to provide access for postoperative care.

Sheep were housed in separate metabolic cages with access to water and food *ad libitum* in a temperature-controlled room (16 ± 1°C, humidity 50 ± 10%) with a 12 h–12 h light–dark cycle. Five days of postoperative recovery was allowed before experiments commenced. During this time, ewes received I.V. antibiotics daily for 4 days (benzylpenicillin sodium, 600 mg; Novaris, Auckland, New Zealand; and gentamycin, 80 mg). Fetal catheters were maintained patent by continuous infusion of heparinized saline (20 IU/ml) at a rate of 0.2 mml/h.

At 104 days, fetuses were randomly allocated to receive an I.V. infusion of normal saline ($n = 6$) or magnesium sulphate heptahydrate dissolved in saline ($MgSO_4.7H_2O$, 500 mg/ml; Phebra, NSW, Australia; $n = 7$) or sham occlusion with saline infusion ($n = 6$). Twenty-four hours before umbilical cord occlusion (UCO), the $MgSO_4$ group received a 160 mg loading dose over 5 min followed by a 48 mg/h maintenance infusion over 24 h before asphyxia (104–105 days) and 24 mg/h for 24 h (105–106 days) after asphyxia, as previously described (Bennet et al., 2018; Galinsky et al., 2015; Galinsky, Dhillon et al., 2018; Galinsky, Draghi et al., 2017). This regimen was adapted from a randomized controlled trial of antenatal $MgSO_4$ and current clinical recommendations (Magpie Trial Follow-Up Study Collaborative Group, 2007).

## Experimental recordings

Fetal mean arterial blood pressure (MAP), corrected for maternal movement by subtraction of amniotic pressure, ECG and EEG were recorded continuously for off-line analysis using custom data-acquisition software (LabView for Windows; National Instruments, TX, USA). The blood pressure signal was collected at 64 Hz and low-pass filtered at 30 Hz. The fetal ECG was analog filtered between 0.05 and 100 Hz, digitized at 512 Hz, and used to derive fetal heart rate. The analog fetal EEG signal was low-pass filtered with a cut-off frequency set with the −3 dB point at 30 Hz, and digitized at a sampling rate of 512 Hz. Power was derived from the power spectrum signal between 0.5 and 20 Hz, and spectral edge was calculated

as the frequency below which 90% of the power was present (Williams & Gluckman, 1990). For data presentation, total EEG power was normalized by logarithmic transformation [in decibels; $10 \times \log(\text{power})$].

## Experimental protocol

Experiments started at 103 days of gestation and ended at 125 days. Fetal MAP, fetal heart rate and EEG were recorded continuously from 2 days before UCO until 21 days (504 h) after occlusion. Fetal asphyxia was induced at 10.00 h by rapid, complete inflation of the umbilical cord occluder for 25 min. Successful occlusion was confirmed by the rapid onset of bradycardia, a rise in MAP and changes in pH and blood gas measurements. Samples of fetal arterial blood were collected at 1 h before occlusion, 5 and 17 min during occlusion, and at 1, 2, 4, 6, 24 (1 day), 168 (7 days) and 504 h (21 days) after the end of occlusion for preductal pH, blood gas (ABL 800; Radiometer, Copenhagen, Denmark) glucose and lactate measurements (model 2300; YSI, OH, USA). Fetal plasma magnesium levels were measured at the baseline (1 h before the start of the I.V. infusion) and 1 h before asphyxia (i.e. 24 h after starting infusion) (Roche/Hitachi 902 clinical chemistry analyser; Hoffman-La Roche, Basel, Switzerland) and were previously published by Galinsky, Dhillon et al. (2018). At the end of the experiment, ewes and fetuses were killed by an overdose of pentobarbitone sodium to the ewe (Pentobarb 300, 9 g; Chemstock International, Christchurch, New Zealand). The rate of fetal loss before the end of the experimental recording period was 20% and did not differ between the groups. In cases of fetal loss, the individual was excluded from study.

## Histopathology

At post-mortem, 21 days after UCO, the fetal brains were perfusion fixed *in situ* with 10% phosphate-buffered formalin. After removal from the skull, tissue was fixed for a further 5 days before processing and embedding using a standard paraffin tissue preparation. Brain slices were cut (10 $\mu$m thick) using a microtome (Leica Jung RM2035; Leica Microsystems, Albany, New Zealand). Brain regions of the forebrain used for analysis included the caudate nucleus (CN) and putamen (PU) at the level of the mid-striatum, and periventricular and intragyral white matter from sections taken 23 mm anterior to stereotaxic zero (Fig. 1). Slides were dewaxed in xylene, rehydrated in decreasing concentrations of ethanol, then washed in 0.1 mol/L PBS. Antigen retrieval was performed in citrate buffer using the pressure cooker technique in an antigen retrieval system (EMS Antigen 200 Retriever; Emgrid, Australia). Endogenous peroxidase quenching was performed by incubation in 0.1% $H_2O_2$ in methanol or

PBS. Non-specific antigens were blocked using 3% normal goat serum. The sections were labelled with 1:200 rabbit anti-NeuN (Abcam; catalogue no. ab177487), 1:200 rabbit anti-Olig-2 (Abcam; catalogue no. ab42453; a marker of oligodendrocytes at all stages of the lineage) (Jakovcevski et al., 2009), 1:200 rabbit anti-Iba1 (Abcam; catalogue no. ab153696), 1:200 rabbit anti-GFAP (Abcam; catalogue no. ab68428), 1:200 mouse anti-MBP (MerkMillipore; catalogue no. MAB381) and mouse anti-adenomatous polyposis coli (CC1; MerkMillipore; catalogue no. OP80-100UG), overnight at 4°C. Sections were incubated in biotin-conjugated IgG (1:200, goat anti-rabbit or anti-mouse; Vector Laboratories, Burlingame, CA, USA) for 3 h at room temperature before being incubated in avidin–biotin complex (Sigma-Aldrich) for 45 min at room temperature or ExtrAvidin (Sigma-Aldrich) at a dilution of 1:200 in 3% normal goat serum for 2 h (for MBP and CC1). Sections were reacted with 3,3′-diaminobenzidine tetrahydrochloride (DAB; Sigma Aldrich). The reaction was stopped by washing in PBS before the slides were dehydrated and mounted.

Myelin (MBP) density and numbers of neurons (NeuN), oligodendrocytes (Olig-2 and CC1), astrocytes (GFAP) and microglia were visualized using light microscopy (Olympus, Tokyo, Japan) at ×40 magnification and cellSens imaging software (v.2.3; Olympus). Positive cells or immunoreactivity were quantified for each region of interest from two sections per animal using ImageJ software (v.2.00; LOCI, University of Wisconsin). NeuN$^+$ cells were counted only if they were morphologically normal; cells displaying condensed or fragmented nuclei were not counted (Pozo Devoto et al., 2006). Microglia (Iba-1$^+$ cells) showing ramified (small cell body with more than one branching process) or amoeboid morphology (large cell bodies, with up to one branching process) were included in our assessment (Galinsky, van

de Looij et al., 2020; Kelly et al., 2021; Nott et al., 2020). The area fraction of GFAP and MBP immunoreactivity was determined with a standard intensity threshold using ImageJ software. Average scores from both hemispheres from two sections were calculated for each region (Fig. 1). All cell counts were performed by assessors who were blinded to the treatment (R.G., S.B.K. and S.K.D.).

## Data analysis and statistics

Off-line physiological data analysis was performed using Labview-based customized programs (Labview for Windows; National instruments). EEG power and spectral edge frequency were processed as hourly averages for analysis and presentation. EEG data are presented from 24 h before UCO until the end of the experiment. EEG power and frequency were normalised by subtracting the baseline average (12 h before starting the MgSO$_4$ or vehicle infusion) from the absolute value. Sleep stage cycling (SSC) was defined using 1 min averaged EEG frequency data from the last 5 h of the experimental period (499–504 h) as a repetitive alternating pattern of high- and low-frequency activity, with each phase lasting ∼20 min. Statistical analysis was undertaken using SPSS (v.22; SPSS, Chicago, IL, USA) and Sigmaplot software (v.12; Systat software, San Jose, CA, USA). Between- and within-group comparisons of fetal blood gases, glucose, lactate and physiological data were performed by two-way repeated-measures ANOVA. Physiological data for the pre- and post-occlusion periods were analysed as separate time periods. When statistical significance was found between groups or between group and time, *post hoc* comparisons were made using Fisher's protected least significant difference test (Levin et al., 1994). Between-group comparisons of fetal weight and brain weight were performed using one-way ANOVA, followed

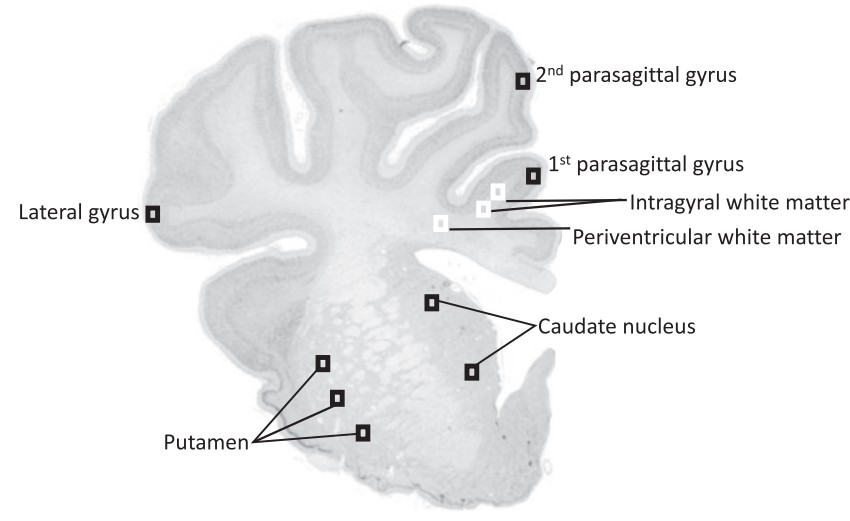

**Figure 1. Schematic diagram indicating fields sampled (regions of interest) for histological assessment**
Areas of the forebrain used for analysis included the premotor cortex, caudate nucleus, putamen and periventricular and intragyral white matter from sections taken 23 mm anterior to stereotaxic zero. Black squares were sampled for assessment of neuronal survival, astrocytes and microglia within the parasagittal and lateral gyri, caudate nucleus and putamen. White squares were sampled for assessment of astrocytes, microglia and oligodendrocytes within the intragyral (IGWM) and periventricular white matter (PvWM).

by Fisher's protected least significant difference test when significance was found between groups. Between-group comparisons of neuropathological data were performed using two-way ANOVA, followed by Fisher's protected least significant difference test when significance was found between groups or between region and group. If there was an effect of region and group, the effect of group was assessed for each region separately. The Mann–Whitney *U*-test was used to compare non-parametric data. Between-group comparisons for numbers of fetuses that developed SSC were performed using Fisher's exact test. Data were tested for normality using the Shapiro–Wilk test. A power analysis for neuronal and oligodendrocyte loss suggested that the study has 90% power to detect a minimum difference of 20 cells/field, with an $\alpha$ of 0.05. Statistical significance was accepted when $P < 0.05$. EEG power and frequency data are presented as hourly means ± SD. Histological assessments are presented as scatter plots with means ± SD.

## Results

### The baseline period

Before the saline or MgSO$_4$ infusions, baseline MAP, fetal heart rate (data not shown) and EEG activity (Fig. 2), blood gases and glucose and lactate concentrations (Table 1) did not differ between groups and were within the normal range by our laboratory standards.

### Physiological changes during baseline MgSO$_4$ infusion before occlusion

In MgSO$_4$-treated fetuses, serum magnesium levels increased to a peak of 1.89 ± 0.08 mmol/L, *vs.* 0.88 ± 0.07 mmol/L in saline control fetuses 1 h before occlusion ($P = 0.0001$; Galinsky, Dhillon et al., 2018). EEG power was reduced before occlusion in the MgSO$_4$ + occlusion group compared with the vehicle + occlusion and sham control groups between −12 and −10, −8 and −6, and −2 and −1 h (12–14, 16–18 and 23–24 h, respectively, after starting MgSO$_4$ infusion; $P = 0.0048$; Fig. 2). Blood gases, acid–base status, glucose and lactate levels did not differ between groups (Table 1).

### Umbilical cord occlusion and recovery

Cardiovascular, cerebrovascular and neurophysiological adaptations before and during asphyxia and in early recovery have been documented in separate studies using a separate cohort of sheep (Galinsky et al., 2015; Galinsky, Dhillon et al., 2018; Galinsky, Draghi et al., 2017). Umbilical cord occlusion was characterised by profound bradycardia and hypotension. The magnitude of bradycardia and hypotension during the final minute of occlusion did not differ between the occlusion groups (fetal heart rate: vehicle + occlusion, 53 ± 3 beats/min *vs.* MgSO$_4$ + occlusion, 53 ± 2 beats/min; MAP: vehicle + occlusion, 11 ± 1 mmHg *vs.* MgSO$_4$ + occlusion, 11 ± 0 mmHg). After release of occlusion, there was rapid recovery of all parameters, similar to previous studies (Galinsky, Draghi et al., 2017; van den Heuij et al., 2019).

After UCO, there was a transient suppression of EEG power and spectral edge frequency that did not differ between groups (Fig. 2). During recovery, EEG power remained suppressed in the occlusion groups compared with sham control animals until the end of the study ($P = 0.0005$; Fig. 2). Spectral edge frequency was suppressed in the occlusion groups compared with sham controls during the first 93 h of recovery ($P = 0.0034$;

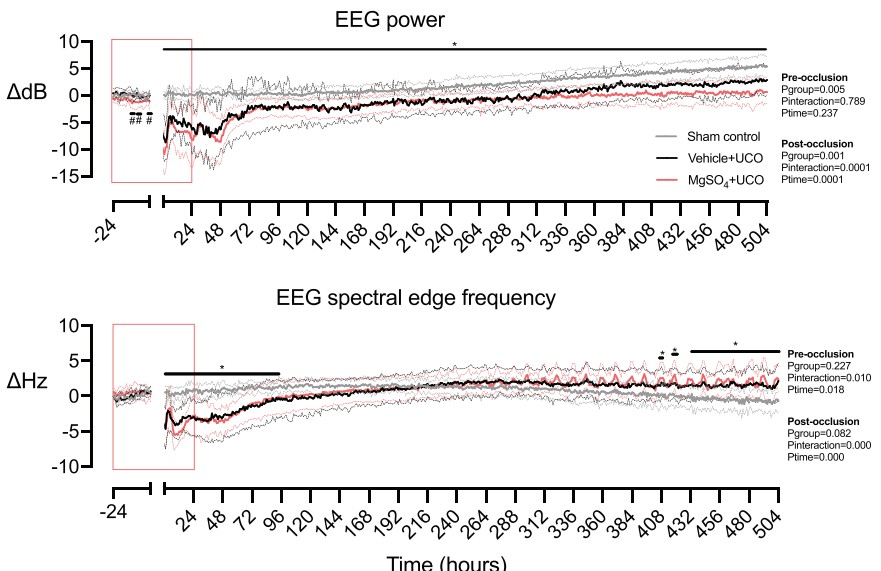

**Figure 2. Time course of EEG power and frequency after umbilical cord occlusion**
EEG power (in decibels) and spectral edge frequency (in herz) in sham control (grey, $n = 6$), vehicle + occlusion (black, $n = 6$) and MgSO$_4$ + occlusion (red, $n = 7$) groups. The red box indicates the period of fetal I.V. saline or MgSO$_4$ infusion. Data are hourly means (continuous lines) ± SD (dashed lines). Data were analysed using a two-way ANOVA, with time as a repeated measure. *Post hoc* comparisons were made using Fisher's protected least significant difference test when significance was found between groups or between groups and time. *$P < 0.05$ occlusion *vs.* sham control, #$P < 0.05$ MgSO$_4$ + occlusion *vs.* vehicle + occlusion and sham control. [Colour figure can be viewed at wileyonlinelibrary.com]

**Table 1. Arterial pH, blood gases, glucose and lactate values**

| Parameter | −60 min | 5 min | 17 min | 1 h | 2 h | 4 h | 6 h | 1 day | 7 days | 14 days | 21 days |
|---|---|---|---|---|---|---|---|---|---|---|---|
| **pH** | | | | | | | | | | | |
| Sham control | 7.36 ± 0.02 | 7.36 ± 0.02 | 7.36 ± 0.02 | 7.36 ± 0.02 | 7.36 ± 0.03 | 7.36 ± 0.02 | 7.36 ± 0.02 | 7.35 ± 0.02 | 7.34 ± 0.02 | 7.35 ± 0.02 | 7.34 ± 0.01 |
| Vehicle + UCO | 7.36 ± 0.03 | 7.03 ± 0.03* | 6.85 ± 0.02* | 7.25 ± 0.04* | 7.26 ± 0.05* | 7.37 ± 0.04 | 7.41 ± 0.01* | 7.38 ± 0.04 | 7.39 ± 0.03 | 7.36 ± 0.02 | 7.36 ± 0.03 |
| MgSO$_4$ + UCO | 7.33 ± 0.02 | 7.03 ± 0.02* | 6.88 ± 0.09* | 7.25 ± 0.02* | 7.31 ± 0.04* | 7.38 ± 0.05 | 7.38 ± 0.04 | 7.35 ± 0.04 | 7.35 ± 0.02 | 7.34 ± 0.02 | 7.36 ± 0.03 |
| **P$_{CO_2}$ (mmHg)** | | | | | | | | | | | |
| Sham control | 48 ± 2 | 47 ± 3 | 46 ± 3 | 46 ± 2 | 47 ± 4 | 44 ± 3 | 46 ± 3 | 49 ± 3 | 51 ± 4 | 49 ± 3 | 51 ± 1 |
| Vehicle + UCO | 49 ± 4 | 101 ± 7* | 135 ± 22* | 47 ± 2 | 48 ± 3 | 49 ± 1* | 48 ± 1 | 45 ± 2 | 46 ± 3 | 48 ± 1 | 48 ± 3 |
| MgSO$_4$ + UCO | 47 ± 3 | 94 ± 6* | 128 ± 10* | 44 ± 6 | 47 ± 6 | 47 ± 6 | 45 ± 5 | 46 ± 4 | 48 ± 3 | 47 ± 4 | 48 ± 4 |
| **P$_{O_2}$ (mmHg)** | | | | | | | | | | | |
| Sham control | 26 ± 2 | 25 ± 3 | 24 ± 2 | 25 ± 3 | 25 ± 3 | 25 ± 2 | 26 ± 2 | 25 ± 3 | 22 ± 3 | 22 ± 3 | 20 ± 4 |
| Vehicle + UCO | 25 ± 2 | 7 ± 2* | 10 ± 3* | 32 ± 3 | 28 ± 4 | 23 ± 4 | 26 ± 4 | 30 ± 3 | 31 ± 4* | 30 ± 4* | 27 ± 4 |
| MgSO$_4$ + UCO | 26 ± 5 | 5 ± 2* | 6 ± 3* | 32 ± 6 | 28 ± 6 | 27 ± 6 | 29 ± 7 | 30 ± 7 | 30 ± 4* | 30 ± 5* | 26 ± 5 |
| **Lactate (mmol/L)** | | | | | | | | | | | |
| Sham control | 0.9 ± 0.2 | 0.9 ± 0.1 | 0.9 ± 0.2 | 0.9 ± 0.1 | 1.0 ± 0.2 | 1.0 ± 0.1 | 1.0 ± 0.1 | 1.0 ± 0.1 | 1.0 ± 0.1 | 1.0 ± 0.3 | 1.1 ± 0.3 |
| Vehicle + UCO | 0.9 ± 0.2 | 4.2 ± 0.5* | 6.1 ± 0.9* | 5.8 ± 1.0* | 5.6 ± 1.5* | 4.0 ± 2.2* | 2.7 ± 1.5 | 2.0 ± 0.8 | 0.8 ± 0.2 | 0.1 ± 0.1 | 0.9 ± 0.1 |
| MgSO$_4$ + UCO | 1.0 ± 0.5 | 3.7 ± 0.7* | 5.9 ± 0.7* | 4.2 ± 1.1*# | 3.6 ± 1.4* | 2.2 ± 1.1 | 2.0 ± 1.5 | 1.7 ± 1.4 | 0.6 ± 0.2 | 0.7 ± 0.2 | 0.8 ± 0.2 |
| **Glucose (mmol/L)** | | | | | | | | | | | |
| Sham control | 1.2 ± 0.3 | 1.0 ± 0.1 | 1.0 ± 0.1 | 1.0 ± 0.1 | 1.2 ± 0.1 | 1.1 ± 0.1 | 1.1 ± 0.1 | 1.2 ± 0.1 | 1.0 ± 0.2 | 0.8 ± 0.2 | 0.7 ± 0.2 |
| Vehicle + UCO | 1.0 ± 0.1 | 0.4 ± 0.1* | 0.6 ± 0.2* | 1.5 ± 0.1* | 1.4 ± 0.1 | 1.4 ± 0.2* | 1.5 ± 0.1* | 1.4 ± 0.4 | 1.0 ± 0.1 | 1.0 ± 0.2 | 0.9 ± 0.1 |
| MgSO$_4$ + UCO | 1.1 ± 0.1 | 0.3 ± 0.1* | 0.6 ± 0.4* | 1.4 ± 0.3* | 1.3 ± 0.2 | 1.4 ± 0.3* | 1.4 ± 0.2* | 1.2 ± 0.3 | 0.9 ± 0.3 | 0.9 ± 0.1 | 0.9 ± 0.2 |

Fetal arterial biochemical values before, during (5 and 17 min) and after (1, 2, 4 and 6 h; 1, 7, 14 and 21 days) umbilical cord occlusion (UCO) in sham occlusion ($n = 6$), vehicle + UCO ($n = 6$) and MgSO$_4$ + UCO ($n = 7$) groups. Data were analysed using two-way ANOVA, followed by Fisher's protected least significant difference *post hoc* comparisons. Data are means ± SD.

* $P < 0.05$ *vs.* sham control.

# $P < 0.05$ *vs.* vehicle + UCO.

Fig. 2). Thereafter, spectral edge frequency reduced progressively in the sham control group compared with both occlusion groups between 430 (122 days of gestation) and 504 h (125 days of gestation; $P = 0.0014$; Fig. 2). The onset of electrographic seizures occurred from ~10 h after occlusion in both occlusion groups (Bennet et al., 2018). Formation of SSC was observed in five of six (86%) fetuses in the sham control group, compared with one of six (17%) in the vehicle + occlusion group ($P = 0.0801$ *vs.* sham control; Fig. 3) and one of seven in the MgSO$_4$ + occlusion group (17%; $P = 0.0291$ *vs.* sham control; Fig. 3).

### Post-mortem findings

There were no differences between groups for body weight and the ratio of males to females (Table 2). Brain weight

**Table 2. Fetal body and brain weight and sex**

| Group | Body weight | Brain weight | Male:female |
|---|---|---|---|
| Sham control | 3.2 ± 0.4 | 40 ± 3 | 4:2 |
| Vehicle + UCO | 3.1 ± 0.4 | 30 ± 4*# | 2:4 |
| MgSO$_4$ + UCO | 3.5 ± 0.2 | 36 ± 3 | 3:4 |

Fetal body weight (kg), brain weight (g) and sex in sham occlusion ($n = 6$), vehicle + umbilical cord occlusion (UCO; $n = 6$) and MgSO$_4$ + UCO ($n = 7$) groups. Data were analysed using one-way ANOVA, followed by Fisher's protected least significant difference *post hoc* comparisons. Data are means ± SD.
*$P < 0.05$ *vs.* sham control.
#$P < 0.05$ *vs.* vehicle + UCO.

was reduced in the vehicle + occlusion group compared with the sham control group ($P = 0.0001$) and the MgSO$_4$ + occlusion group ($P = 0.005$).

### Histopathology

**Premotor cortex and basal ganglia.** The numbers of GFAP$^+$ cells and area fraction of GFAP$^+$ staining were increased in the vehicle + occlusion group compared with the sham control group [GFAP$^+$ cells, $P = 0.0003$ (CN) and 0.0003 (PU); GFAP area fraction, $P = 0.0032$ (cortex) and 0.0019 (PU); Figs. 4 and 5]. In the MgSO$_4$ + occlusion group, numbers of GFAP$^+$ cells and area fraction of GFAP$^+$ staining were reduced compared with the vehicle + occlusion group [GFAP$^+$ cells, $P = 0.0006$ (cortex), 0.0004 (CN) and 0.0021 (PU); GFAP area fraction, $P = 0.001$ (cortex), 0.0167 (CN) and 0.0004 (PU); Figs. 4 and 5]. In the vehicle + occlusion group, total numbers of Iba-1$^+$ microglia were increased in the caudate nucleus compared with the sham control group ($P = 0.0051$). In the MgSO$_4$ + occlusion group, numbers of Iba-1$^+$ microglia were reduced in the caudate nucleus compared with the vehicle + occlusion group ($P = 0.0436$). In the cerebral cortex and putamen, numbers of Iba-1$^+$ microglia did not differ between groups. Numbers of amoeboid microglia were increased in caudate nucleus in both occlusion groups compared with the sham control group ($P_{vehicle+UCO} = 0.0450$ and $P_{MgSO4+UCO} = 0.0462$). There were no significant differences in numbers of amoeboid microglia between groups within the cortex and putamen ($P = 0.0647$ vehicle + occlusion *vs.* MgSO$_4$ + occlusion; Figs. 4 and 5). The number of NeuN$^+$ cells was reduced in the cerebral cortex ($P_{vehicle+UCO} = 0.0141$ and $P_{MgSO4+UCO} = 0.0089$), caudate nucleus ($P_{vehicle+UCO} = 0.0001$ and $P_{MgSO4+UCO} = 0.0015$) and putamen ($P_{vehicle+UCO} = 0.0020$ and $P_{MgSO4+UCO} = 0.0011$) in both occlusion groups compared with the sham control group (Figs. 4 and 5). There were

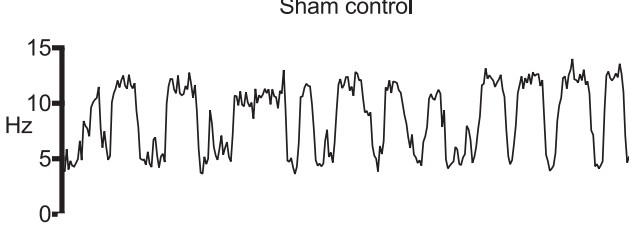
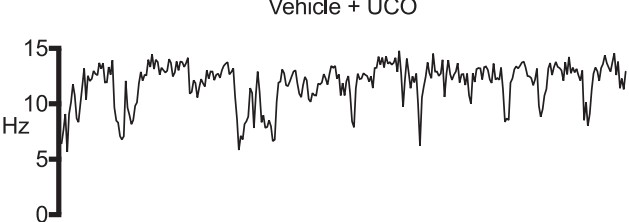
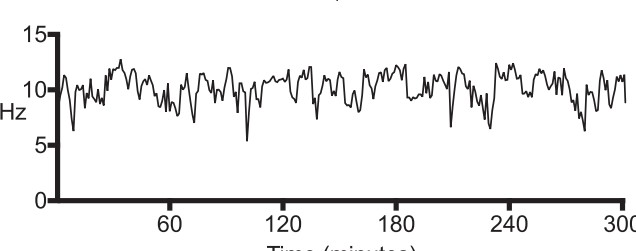

**Figure 3. Examples of spectral edge frequency at the end of the experiment**
Representative examples of continuous EEG data (spectral edge frequency, 1 min averages) from the last 5 h of the experimental period (499–504 h), showing distinct periods of high- and low-frequency activity characteristic of sleep stage cycling (SSC) in a fetus from the sham control group (top). The lack of continuous cycling of high-and low-frequency activity in the vehicle + occlusion (middle) and MgSO$_4$ + occlusion (bottom) groups indicated impaired development of SSC. Numbers of fetuses in each group that developed SSC were compared using Fisher's exact test.

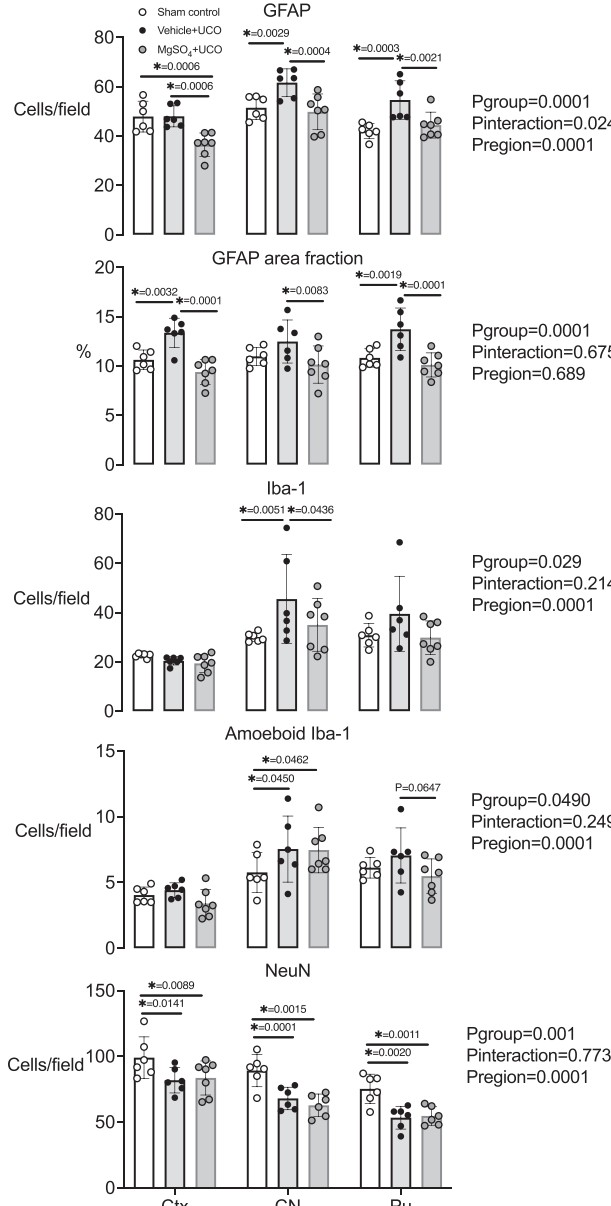

no differences in the number of surviving NeuN$^+$ cells between the saline + occlusion and MgSO$_4$ + occlusion groups in the cerebral cortex and striatum.

**Periventricular and intragyral white matter tracts.** In the intragyral white matter, numbers of GFAP$^+$ cells were increased in the vehicle + occlusion group compared with the sham control group ($P = 0.0330$). There were no differences in the numbers of GFAP$^+$ cells between the occlusion groups. In periventricular white matter, the numbers of GFAP$^+$ cells were increased in the vehicle + occlusion group compared with the sham control group ($P = 0.0061$). In the MgSO$_4$ + occlusion group, numbers of GFAP$^+$ cells were lower compared with the vehicle + occlusion group ($P = 0.0324$). In the intragyral and periventricular white matter tracts (IGWM and PvWM, respectively), the area fraction of GFAP$^+$ staining was reduced in the MgSO$_4$ + occlusion group compared with the vehicle + occlusion group [$P = 0.0015$ (IGWM) and 0.0390 (PvWM); Figs. 6 and 7].

In the periventricular white matter, numbers of Iba-1$^+$ microglia were not significantly increased in the vehicle + occlusion group compared with the sham control group ($P = 0.0545$). In the MgSO$_4$ + occlusion group, numbers of Iba-1$^+$ microglia were reduced compared with the vehicle + occlusion group in the periventricular

**Figure 4. Cell counts and area fractions in the premotor cortex, caudate nucleus and putamen**

Numbers and area fraction of glial fibrillary acidic protein (GFAP)$^+$ staining, and numbers of ionized calcium binding adapter molecule-1 (Iba-1), ameboid Iba-1 and neuronal nuclei (NeuN) cell counts in the premotor cortex (Ctx), caudate nucleus (CN) and putamen (Pu) in sham control (white circles, *n* = 6), vehicle + umbilical cord occlusion (UCO) (black circles, *n* = 6) and MgSO$_4$ + UCO (grey circles, *n* = 7) groups. Data are scatter plots showing the means and SD. Data were analysed using a two-way ANOVA, with treatment as an independent factor and brain region as a repeated measure. *Post hoc* comparisons were made using Fisher's protected least significant difference test. *$P < 0.05$

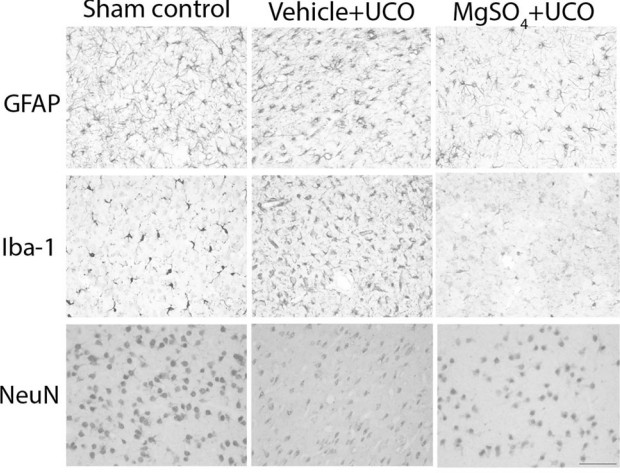

**Figure 5. Photomicrographs of astrocytes, microglia and neurons from the caudate nucleus**

Photomicrographs showing representative examples of glial fibrillary acidic protein (GFAP), ionized calcium binding adapter molecule-1 (Iba-1) and neuronal nuclei (NeuN) staining in the caudate nucleus. Abbreviation: UCO, umbilical cord occlusion. Scale bar: 50 $\mu$m.

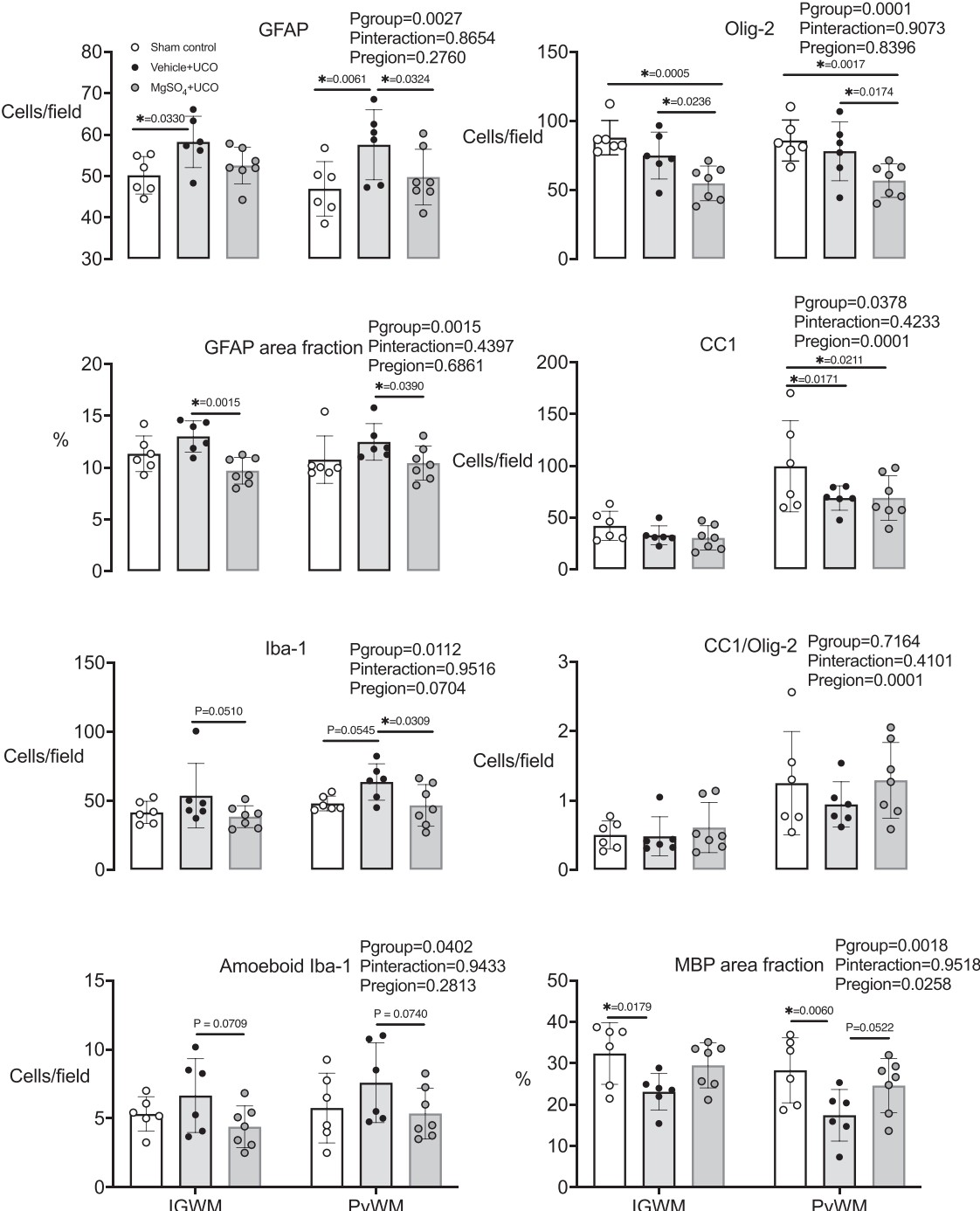

**Figure 6. Cell counts and area fractions in white matter**

Numbers and area fraction of glial fibrillary acidic protein (GFAP)[+] staining, and numbers of ionized calcium binding adapter molecule-1 (Iba-1), ameboid Iba-1, oligodendrocyte transcription factor-2 (olig-2), anti-adenomatous poly-posis coli clone (CC1)[+] cells, the ratio of CC1[+] to Olig2[+] cells and the area fraction of myelin basic protein (MBP)[+] staining in the intragyral and periventricular white matter (IGWM and PvWM, respectively) in sham control (open circles, $n = 6$), vehicle + umbilical cord occlusion (UCO; black circles, $n = 6$) and MgSO$_4$ + UCO (grey circles, $n = 7$) groups. Data are scatter plots showing means and SD. Data were analysed using a two-way ANOVA, with treatment as an independent factor and brain region as a repeated measure. *Post hoc* comparisons were made using Fisher's protected least significant difference test. *$P < 0.05$

white matter ($P = 0.0309$). In the intragyral and periventricular white matter tracts, numbers of amoeboid microglia were not significantly lower in the MgSO$_4$ + occlusion group compared with the vehicle + occlusion group ($P = 0.07$ vehicle + occlusion *vs.* MgSO$_4$ + occlusion for both regions; Figs. 6 and 7). Numbers of Olig-2$^+$ oligodendrocytes did not differ between vehicle + occlusion and sham control groups in the intragyral and periventricular white matter tracts, whereas the MgSO$_4$ + occlusion group was associated with fewer Olig-2$^+$ oligodendrocytes compared with the vehicle + occlusion group [$P = 0.0236$ (IGWM) and $0.0174$ (PvWM)] and the sham control group [$P = 0.0005$ (IGWM) and $0.0017$ (PvWM)].

Numbers of mature CC1$^+$ oligodendrocytes were lower in both occlusion groups compared with the sham occlusion group in the periventricular white matter ($P_{\text{vehicle+UCO}} = 0.0171$ and $P_{\text{MgSO4+UCO}} = 0.0211$; Figs. 6 and 7). The ratio of CC1$^+$ cells (mature oligodendrocytes) to Olig-2$^+$ cells (total oligodendrocytes) did not differ

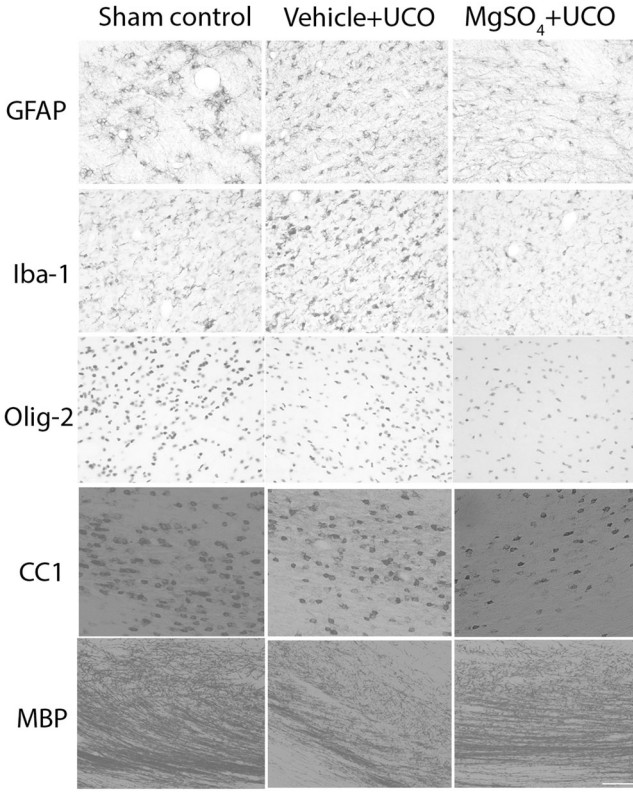

**Figure 7. Photomicrographs of astrocytes, microglia and oligodendrocytes in periventricular white matter**
Photomicrographs showing representative glial fibrillary acidic protein (GFAP), ionized calcium binding adapter molecule-1 (Iba-1), oligodendrocyte transcription factor (olig-2), anti-adenomatous polyposis coli clone (CC1) and myelin basic protein (MBP) staining in the periventricular white matter. Abbreviation: UCO, umbilical cord occlusion. Scale bar: 50 $\mu$m.

between groups. In the intragyral and periventricular white matter tracts, the area fraction of MBP$^+$ staining was reduced in the vehicle + occlusion group compared with the sham control group [$P = 0.0179$ (IGWM) and $0.0060$ (PvWM); Figs. 6 and 7]. In the MgSO$_4$ + occlusion group, the area fraction of MBP$^+$ staining did not differ from the sham occlusion or vehicle + occlusion group (MgSO$_4$ + occlusion *vs.* vehicle + occlusion, $P = 0.0826$ and $0.0522$ for the intragyral and periventricular white matter tracts, respectively).

## Discussion

This study demonstrates that a clinically comparable increase in fetal plasma magnesium concentration for 24 h before and after profound hypoxia-ischaemia in preterm fetal sheep did not improve maturation of EEG power or frequency after recovery to term-equivalent age, 21 days later. Furthermore, MgSO$_4$ did not improve survival of cortical and subcortical neurons and was associated with greater overall loss of oligodendrocytes compared with the vehicle-treated group. Interestingly, MgSO$_4$ was associated with reduced grey and white matter gliosis and a regional, intermediate improvement in myelin density compared with vehicle.

In the present study, the fetal MgSO$_4$ infusion achieved comparable serum magnesium levels to those seen clinically with current guidelines (Boriboonhirunsarn et al., 2012; Borja-Del-Rosario et al., 2014; McGuinness et al., 1980). From ∼10 h after UCO, we observed large-amplitude stereotypical seizures (Bennet et al., 2018; Galinsky, Draghi et al., 2017), similar to previous studies (Bennet et al., 2007, 2018; Drury et al., 2013; Drury, Davidson, Bennet et al., 2014; Drury, Davidson, Mathai et al., 2014; Galinsky, Draghi et al., 2017; Koome et al., 2013). These stereotypical evolving seizures are preceded by secondary loss of mitochondrial function (Bennet et al., 2006; Gonzalez et al., 2005) and, clinically, are strongly associated with adverse neurological outcome (Glass, 2014). Magnesium is a key physiological inhibitor of neural glutamatergic activity (Zeevalk & Nicklas, 1992). Using the same experimental model and treatment protocol, we have previously shown that MgSO$_4$ infusion was associated with a marked reduction in the numbers of seizures and seizure burden, strongly supporting a significant central effect on the NMDA receptor (Bennet et al., 2018; Galinsky, Draghi et al., 2017). Thus, pragmatically, these data imply that the infusion protocol was sufficient to provide a significant central anti-excitatory effect.

During recovery from UCO there was a sustained reduction in EEG power and frequency (Bennet et al., 2006; Dean et al., 2006; Hunter et al., 2003). This suppression was not affected by MgSO$_4$ infusion, and

EEG activity recovered progressively towards baseline levels after 48−72 h. Suppression of EEG activity in the latent phase after asphyxia is an active process mediated by inhibitory neuromodulators, such as adenosine (Hunter et al., 2003; Von Lubitz et al., 1999) and neurosteroids (Hirst et al., 2008; Yawno et al., 2007). The present observation that $MgSO_4$ treatment did not modulate this background EEG suppression after asphyxia is consistent with previous data showing no effect of antenatal or postnatal $MgSO_4$ treatment on amplitude-integrated or continuous EEG activity in asphyxiated preterm fetal sheep (Galinsky, Draghi et al., 2017), near term/term human neonates and term neonatal piglet studies (Groenendaal et al., 2002; Levene et al., 1995; Lingam et al., 2019). We now show that $MgSO_4$ infusion did not improve maturation of EEG power or frequency for 21 days after asphyxia, with significantly lower EEG power and higher spectral edge frequency (from days 17 to 21) in both occlusion groups compared with the sham control group. The higher spectral edge frequency at this age is consistent with the impaired maturation of SSC observed in both occlusion groups compared with sham occlusion, probably attributable to loss of inhibitory GABAergic neurons within the cortical and deep grey matter (Hassani et al., 2009; Mahon et al., 2006; Qiu et al., 2010). Collectively, these data suggest that the anti-excitatory effects of $MgSO_4$ do not improve functional maturation of EEG activity.

Pathologically, the model of preterm HIE presented here is consistent with the common clinical pattern of impaired oligodendrocyte maturation and diffuse white matter loss, with evolving loss of cortical and subcortical grey matter (Buser et al., 2012; Huang & Castillo, 2008; Lear et al., 2021, 2022; Riddle et al., 2011; van den Heuij et al., 2019). Despite the significant anti-excitatory effects of $MgSO_4$, in the present study $MgSO_4$ was not associated with improved survival of cortical or basal ganglia neurons after severe asphyxia. These data are consistent with previous reports of a lack of effect of antenatal $MgSO_4$ treatment on neuronal damage after asphyxia in preterm (Galinsky, Draghi et al., 2017) and term-equivalent fetal sheep (de Haan et al., 1997), and after hypoxia–ischaemia in newborn piglets (Greenwood et al., 2000; Penrice et al., 1997). These findings support the concept that electrographic seizure activity during the secondary phase of recovery from an asphyxial insult is primarily a marker of neural injury rather than a substantial cause of secondary neuronal loss.

In contrast, $MgSO_4$ was associated with reduced astrocytosis in the motor cortex and basal ganglia and reduced numbers of microglia in the caudate nucleus compared with vehicle + occlusion. However, numbers of amoeboid microglia were not different between the occlusion groups. Likewise, in the intragyral and periventricular white matter tracts, $MgSO_4$

reduced astrogliosis and numbers of microglia but did not significantly reduce microglial activation ($P = 0.07$, $MgSO_4$ + occlusion *vs.* vehicle + occlusion). The exact mechanisms underpinning this observation are unclear, but $MgSO_4$ has anti-inflammatory effects linked to direct modulation of nuclear factor-$\kappa$B signalling and inhibition of L-type calcium channels (Lin et al., 2010; Sugimoto et al., 2012). In the present study, we administered $MgSO_4$ directly to the fetus, because $Mg^{2+}$ transfer across the sheep placenta is limited (Akoury et al., 1997). It is reasonable to expect that direct infusion to the fetus must also expose the placenta to $MgSO_4$, although this was not assessed directly in our study. This is important, because $MgSO_4$ has been associated with anti-inflammatory effects in the placenta, circulation and CNS (Daher et al., 2018; Khatib et al., 2022; Sugimoto et al., 2012). It is well established that inflammation is an important extrinsic mechanism involved in the activation of pro-apoptotic pathways during the evolution of HIE (Mallard et al., 2014). The present observations suggest, encouragingly, that $MgSO_4$ is at least moderately effective at modulating persistent grey and white matter inflammation after HI.

Conversely, it is important to note that two subgroup analyses of infants exposed to clinical chorioamnionitis from a large randomized controlled trial (Rouse et al., 2008) showed that $MgSO_4$ was not associated with improved neurodevelopment at 2 years of age or with reduced rates of intraventricular haemorrhage or periventricular leucomalacia (Edwards et al., 2018; Kamyar et al., 2016). Although both data sets were relatively small and could be confounded by variability in the duration of $MgSO_4$ infusion relative to the timing of delivery (i.e. limiting transfer of $MgSO_4$ to the fetus), they suggest that $MgSO_4$ did not modulate neurodevelopmental impairments associated with exposure to antenatal inflammation.

In vehicle-treated fetuses in the present study, occlusion was associated with no change in total numbers of (Olig-$2^+$) oligodendrocytes in the intragyral and periventricular white matter tracts, probably reflecting restorative proliferation (Lear et al., 2021, 2022; van den Heuij et al., 2019). In contrast, $MgSO_4$ treatment was associated with reduced total numbers of (Olig-$2^+$) oligodendrocytes after occlusion ($P < 0.05$ $MgSO_4$ + occlusion *vs.* sham control and vehicle + occlusion groups). Interestingly, we reported previously that in the same model, $MgSO_4$ was associated with a greater reduction in total oligodendrocytes 3 days after occlusion, but increased cell proliferation compared with vehicle + occlusion (Galinsky, Draghi et al., 2017). Furthermore, there was no difference in the proportion of immature and mature (CNPase$^+$) oligodendrocytes between groups 3 days after occlusion, suggesting that the greater reduction in oligodendrocyte numbers in

$MgSO_4$-treated fetuses was not unique to a specific stage of oligodendrocyte development (Galinsky, Draghi et al., 2017). However, the white matter tracts of the fetal sheep brain at 0.7 of gestation predominately include immature pre-myelinating oligodendrocytes, similar to the preterm human brain at 28–30 weeks of gestation (Back, 2017; Galinsky, Dhillon et al., 2020; Galinsky, Draghi et al., 2017; Galinsky, van de Looij et al., 2020). We now show that 21 days after asphyxia, at a time when myelination in the fetal sheep brain is comparable to the term human brain, despite the greater loss of total (Olig-2$^+$) oligodendrocytes in the $MgSO_4$ + occlusion group compared with vehicle + occlusion, there was a similar reduction in the numbers of mature, myelinating (CC1$^+$) oligodendrocytes in the periventricular white matter of both occlusion groups (Lear et al., 2021, 2022; van den Heuij et al., 2019).

The specific mechanism responsible for the overall loss of Olig-2$^+$ oligodendrocytes in the $MgSO_4$ group is not known. It is well established that NMDA receptors are present on oligodendrocytes and are activated during hypoxia–ischaemia (Karadottir et al., 2005). Both immature and mature oligodendrocytes exhibit glutamate-evoked currents, which can be inhibited by magnesium (Karadottir et al., 2005). Indeed, neurons send synaptic inputs to oligodendrocytes residing within cerebral grey and white matter structures (Karadottir et al., 2005; Kukley et al., 2007). This form of neuronal–oligodendrocyte signalling might contribute to oligodendrocyte differentiation and stimulation of axonal myelination (Kolodziejczyk et al., 2010). The lack of long-term benefit to total oligodendrocyte survival from $MgSO_4$ treatment suggests that prolonged NMDA glutamate receptor blockade by magnesium cannot salvage the acute loss of oligodendrocytes after severe asphyxia.

Multiple studies have demonstrated that recovery from hypoxia–ischaemia in preterm fetal sheep and human infants is associated with initial loss of immature oligodendrocytes followed by intense proliferation of oligodendrocyte progenitors that results in restoration of total cell numbers, but impaired lineage maturation and reduced myelination (Buser et al., 2012; Drury, Davidson, Bennet et al., 2014; van den Heuij et al., 2019). In contrast, in the present study we demonstrate that the greater reduction in the total numbers of oligodendrocytes observed in $MgSO_4$-treated fetuses during secondary energy failure (3 days after hypoxia–ischaemia) does not reduce numbers of mature myelinating oligodendrocytes during the tertiary phase after hypoxia–ischaemia (21 days after the insult).

Unexpectedly, despite the greater reduction in total (Olig-2$^+$) oligodendrocytes and a similar reduction in mature (CC1$^+$) oligodendrocytes after $MgSO_4$ infusion compared with vehicle, there was an intermediate improvement in MBP$^+$ myelin density in $MgSO_4$-treated fetuses. Consistent with these data, Koning et al. (2018) showed a reduction in MBP$^+$ tissue loss 7 days after neonatal rats underwent HI on postnatal day 4. However, the serum $Mg^{2+}$ concentration achieved by Koning et al. (2018; 2.7–4.1 mmol/L) was greater than the serum $Mg^{2+}$ levels achieved in the present study (1.89 mmol/L), which is comparable to levels in cord blood samples from preterm human neonates after antenatal $MgSO_4$ treatment (Boriboonhirunsarn et al., 2012; Borja-Del-Rosario et al., 2014; McGuinness et al., 1980). The exact mechanisms underpinning this modest $MgSO_4$-induced improvement in myelin density are unclear. However, we might reasonably speculate that the anti-excitotoxic and anti-inflammatory effects of $MgSO_4$ demonstrated here and in other studies (Daher et al., 2018; Lingam et al., 2019) might have improved the milieu for myelin deposition by surviving oligodendrocytes. Alternatively, it is possible that surviving oligodendrocytes had improved myelination capacity, as shown by a study in adult mice that reported improved myelination during motor learning by surviving oligodendrocytes after a demyelinating injury (Bacmeister et al., 2020).

To the best of our knowledge, this is the first study in a large animal translational model of preterm hypoxia ischaemia to evaluate the impact of $MgSO_4$ on EEG maturation, tertiary gliosis and white and grey matter cell survival after 21 days of recovery to term-equivalent age. Likewise, in neonatal rats exposed to hypoxia–ischaemia on postnatal day 4 (Daher et al., 2018), after 40 days of recovery, postnatal $MgSO_4$ treatment was associated with reduced thalamic and hippocampal tissue loss. However, the $MgSO_4$-induced reduction in tissue loss was only observed in males and was not associated with significant improvements in motor function or cognition (Daher et al., 2018).

In conclusion, administration of $MgSO_4$ to preterm fetal sheep before and after hypoxia–ischaemia did not improve subsequent maturation of EEG activity or promote neuronal or oligodendrocyte survival within the cortical and deep grey matter after 21 days of recovery. Indeed, of concern, $MgSO_4$ was associated with reduced total numbers of oligodendrocytes in white matter tracts and did not improve survival of mature myelinating oligodendrocytes. Interestingly, despite this, $MgSO_4$ infusion reduced white and grey matter gliosis and was associated with a localised, partial improvement in myelin density. The combination of persistent suppression of total (Olig-2$^+$) oligodendrocytes many weeks after exposure to $MgSO_4$, but improved myelination, is unexpected but intriguing. Speculatively, it might be mediated indirectly, through reduced maturational inhibition by gliosis or improved function of the remaining mature

oligodendrocytes (Back, 2017). Further investigations to understand the mechanisms underpinning the intermediate improvement in myelination with MgSO$_4$ treatment are essential.

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

## Additional information

### Data availability statement

Datasets used during the current study are available from the corresponding author upon reasonable request.

### Competing interests

None.

### Author contributions

R.G., L.B. and A.J.G. conceptualized and designed the study. R.G., S.K.D., S.B.K., G.W., J.D., C.A.L. and L.G.v.d.H. undertook experiments and analysed data. R.G., S.B.K. and S.K.D. undertook immunohistochemistry, cell quantification, analysis and preparation of figures. All authors critically reviewed and approved the final manuscript and agree to be accountable for all aspects of the work in ensuring that questions related to the accuracy or integrity of any part of the work are appropriately investigated and resolved. All persons designated as authors qualify for authorship, and all those who qualify for authorship are listed.

### Funding

This study was supported by the Health Research Council of New Zealand (grants 17/601 and 22/559), the Auckland Medical Research Foundation, the Lottery Health Grants Board of New Zealand, the C. J. Martin Postdoctoral Fellowship and project grant from the National Health and Medical Research Council of Australia (APP1090890 and APP1164954) and the Victorian Government's Operational Infrastructure Support Program.

## Acknowledgements

The authors gratefully acknowledge the technical assistance of Mrs Rani Wilson and Mr Vaho Maisashvilli.

Open access publishing facilitated by The University of Auckland, as part of the Wiley - The University of Auckland agreement via the Council of Australian University Librarians.

## Keywords

asphyxia, brain, magnesium sulphate, neuroprotection, preterm birth

## Supporting information

Additional supporting information can be found online in the Supporting Information section at the end of the HTML view of the article. Supporting information files available:

**Statistical Summary Document**
**Peer Review History**

