## [Peer Review History · The Journal of Physiology]

MgSO₄ reduces tertiary gliosis but does not improve EEG recovery, or white or grey matter cell survival after asphyxia in preterm fetal sheep

Robert Galinsky, Simerdeep Kaur Dhillon, Sharmony B Kelly, Guido Wassink, Joanne Davidson, Christopher Arthur Lear, Lotte G. van den Heuij, Laura Bennet, and Alistair J. Gunn
DOI: 10.1113/JP284381

Corresponding author(s): Robert Galinsky (robert.galinsky@hudson.org.au)

The following individual(s) involved in review of this submission have agreed to reveal their identity: Barbara S. Stonestreet (Referee #1)

Review Timeline:

Submission Date:	15-Jan-2023
Editorial Decision:	03-Feb-2023
Revision Received:	21-Feb-2023
Editorial Decision:	08-Mar-2023
Revision Received:	09-Mar-2023
Accepted:	09-Mar-2023

Senior Editor: Harold Schultz

Reviewing Editor: Janna Morrison

Transaction Report:

Dear Dr Galinsky,

Re: JP-RP-2023-284381 "MgSO₄ reduces tertiary gliosis but does not improve EEG recovery, or white or grey matter cell survival after asphyxia in preterm fetal sheep" by Robert Galinsky, Simerdeep Kaur Dhillon, Sharmony B Kelly, Guido Wassink, Joanne Davidson, Christopher Arthur Lear, Lotte G. van den Heuij, Laura Bennet, and Alistair J. Gunn

Thank you for submitting your manuscript to The Journal of Physiology. It has been assessed by a Reviewing Editor and by 2 expert referees and we are pleased to tell you that it is acceptable for publication following satisfactory revision.

REVISION CHECKLIST:

We look forward to receiving your revised submission.

Yours sincerely,

Harold D Schultz
Senior Editor
The Journal of Physiology
<https://jp.msubmit.net>
<http://jp.physoc.org>
The Physiological Society
Hodgkin Huxley House
30 Farringdon Lane
London, EC1R 3AW
UK
<http://www.physoc.org>
<http://journals.physoc.org>

REQUIRED ITEMS FOR REVISION

- Author photo and profile. First (or joint first) authors are asked to provide a short biography (no more than 100 words for one author or 150 words in total for joint first authors) and a portrait photograph. These should be uploaded and clearly labelled with the revised version of the manuscript. See Information for Authors for further details.
 - Please upload separate high-quality figure files via the submission form.
 - Your paper contains Supporting Information of a type that we no longer publish. Any information essential to an understanding of the paper must be included as part of the main manuscript and figures. The only Supporting Information that we publish are video and audio, 3D structures, program codes and large data files. Your revised paper will be returned to you if it does not adhere to our Supporting Information Guidelines
 - A Statistical Summary Document, summarising the statistics presented in the manuscript, is required upon revision. It must be on the Journal's template, which can be downloaded from the link in the Statistical Summary Document section here: https://jp.msubmit.net/cgi-bin/main.plex?form_type=display_requirements#statistics
 - Papers must comply with the Statistics Policy https://jp.msubmit.net/cgi-bin/main.plex?form_type=display_requirements#statistics
- In summary:
- If n {less than or equal to} 30, all data points must be plotted in the figure in a way that reveals their range and distribution. A bar graph with data points overlaid, a box and whisker plot or a violin plot (preferably with data points included) are acceptable formats.
 - If $n > 30$, then the entire raw dataset must be made available either as supporting information, or hosted on a not-for-profit repository e.g. FigShare, with access details provided in the manuscript.
 - 'n' clearly defined (e.g. x cells from y slices in z animals) in the Methods. Authors should be mindful of pseudoreplication.
 - All relevant 'n' values must be clearly stated in the main text, figures and tables, and the Statistical Summary Document (required upon revision)

-The most appropriate summary statistic (e.g. mean or median and standard deviation) must be used. Standard Error of the Mean (SEM) alone is not permitted.

-Exact p values must be stated. Authors must not use 'greater than' or 'less than'. Exact p values must be stated to three significant figures even when 'no statistical significance' is claimed.

-Statistics Summary Document completed appropriately upon revision

-A Data Availability Statement is required for all papers reporting original data. This must be in the Additional Information section of the manuscript itself. It must have the paragraph heading "Data Availability Statement". All data supporting the results in the paper must be either: in the paper itself; uploaded as Supporting Information for Online Publication; or archived in an appropriate public repository. The statement needs to describe the availability or the absence of shared data. Authors must include in their Statement: a link to the repository they have used, or a statement that it is available as Supporting Information; reference the data in the appropriate sections(s) of their manuscript; and cite the data they have shared in the References section. Whenever possible the scripts and other artefacts used to generate the analyses presented in the paper should also be publicly archived. If sharing data compromises ethical standards or legal requirements then authors are not expected to share it, but must note this in their Statement. For more information, see our Statistics Policy.

-Please include an Abstract Figure file, as well as the figure legend text within the main article file. The Abstract Figure is a piece of artwork designed to give readers an immediate understanding of the research and should summarise the main conclusions. If possible, the image should be easily 'readable' from left to right or top to bottom. It should show the physiological relevance of the manuscript so readers can assess the importance and content of its findings. Abstract Figures should not merely recapitulate other figures in the manuscript. Please try to keep the diagram as simple as possible and without superfluous information that may distract from the main conclusion(s). Abstract Figures must be provided by authors no later than the revised manuscript stage and should be uploaded as a separate file during online submission labelled as File Type 'Abstract Figure'. Please ensure that you include the figure legend in the main article file. All Abstract Figures should be created using BioRender. Authors should use The Journal's premium BioRender account to export high-resolution images. Details on how to use and access the premium account are included as part of this email.

EDITOR COMMENTS

Reviewing Editor:

This study investigates the role of MgSO₄ on brain development. Interestingly, although MgSO₄ improves EEG outcomes there is still loss of neurons. But this is an important study in understanding the underlying mechanisms for the potential benefits of such treatment.

The key points section may be over the word limit.

Page 4 - Should 'neuroprotective' be 'neuroprotection'?

Page 4 - It would be helpful to mention glutamate in this paragraph. It comes into the discussion but associating it with NMDA here would improve clarity. It was confusing to read 'anti-excitatory'. Could rephrasing to something like '.. Mg inhibits the excitatory NT glutamate by inhibiting NMDA'?

Page 6 - Were there nineteen animals in the study (6, 7, 6)?

Page 8 - It is stated that studies occurred between 103-125. Does this mean that day one could have been anywhere between 103-125 with tissue collection 21d later (124-146)? Or does this mean that every study started at d 103 and ended at d 125? If the former, this is a very wide range and brain development would be very different at tissue collection. If the later, please clarify.

Page 14 - JP does not have supplementary data.

Page 17 - Please remove 'strongly' from before infer.

It would be helpful to use line numbers.

Please include the sample size in the figure legends.

Senior Editor:

Comments for Authors to ensure the paper complies with the Statistics Policy:

Please state actual p values throughout including figures. Please include sample size(s) and statistical test(s) used in figure legends.

Comments to the Author:

The reviewers have found merit in the study, but the manuscript will require revision based on their comments. Please state actual p values throughout including figures and include in figure legends the sample size(s) and statistical test(s) used in the figure.

REFeree COMMENTS

Referee #1:

General Comments

This is a very well-done study in a complex model. It is meritorious that the investigators are able to maintain the fetal sheep after severe umbilical cord occlusion for 21 days. The model has the advantage that it more closely represents the human pregnancy and human brain than rodent models and physiological determinations can be quantified during the studies.

Magnesium sulfate is now widely used to reduce the development of cerebral palsy in women at risk for early preterm birth in countries worldwide. The authors have examined the ability of magnesium sulfate (MgSO₄) to protect the immature fetal brain after exposure to severe umbilical cord occlusion. They have extended their previous work that examined short term outcomes after neuroprotection with MgSO₄. In the current study they have examined the fetus over a prolonged recovery period of 21 days. They have used appropriate doses of MgSO₄ which closely simulate those used to treat pregnant women before preterm birth. Physiological measures including EEG were performed over the 21 day period of study. Control animals were included. The post-mortem brains were obtained for analysis 21 days after umbilical cord occlusion. They found that MgSO₄ did not improve long-term EEG recovery. However, MgSO₄ infusion attenuated post-occlusion astrocytosis (GFAP+) and microgliosis in the premotor cortex and striatum but did not affect reactive (ameboid) microglia or improve neuronal survival. MgSO₄ was associated with fewer total (Olig-2+) oligodendrocytes in the periventricular and intragyral white matter. The numbers of mature oligodendrocytes were also reduced in the occlusion groups compared to the sham occlusion. MgSO₄ also improved myelin density in the intragyral and periventricular white matter tracts. Consequently, MgSO₄ infusion had only modest effects on white matter in the immature fetal sheep after umbilical cord occlusion but did not contribute to neuronal sparing.

Since MgSO₄ is widely used to treat pregnant women with threatened preterm labor, understanding the potential mechanisms underlying its effect is translationally relevant. The results of the study support the authors conclusions. The discussion is well written and addresses all of the important aspects of the data. There are only a few points that need to be considered.

Specific Comments

1. Page 4, line 20: 'Neuroprotection' not 'neuroprotective'.
2. Page 7, line 12. Presumably, leads and catheters were exteriorized.

3. Page 17, line 18. Where is the data to support the conclusion that MgSO₄ reduced total seizures in this study. This reviewer searched for the data and was unable to locate it. Please provide the data to support this conclusion in the current study.

4. Did they not experience any fetal loss during the studies?

Referee #2:

This work by Galinsky and colleagues examines whether antenatal administration of MgSO₄ provides longer term neuroprotective benefit with the preterm brain, assessed using histological analyses combined with EEG outcomes. A comprehensive set of data are presented for grey and white matter brains regions, with results broadly showing that MgSO₄ did not improve brain activity after severe asphyxial insult, and while MgSO₄ modified the neuroinflammatory response in a region-specific manner, it was associated with cell loss of neurons and oligodendrocytes.

Overall this is an important and timely study, and the results are likely to be highly interpretable to the clinical situation.

There are however a few points that require revision:

I found myself reading the Introduction a couple of times in an attempt for clarity on the rationale for the study. The way it is currently written, the study appears to be incremental given that you have previously demonstrated a lack of protection in preterm sheep with MgSO₄ in brains collected at 3 days. However I think this current study has merit to fill a knowledge gap on the effects of preterm exposure to MgSO₄ at term-equivalent brain age.

Some questions that you might consider for clarity:

In the Doyle 2009 meta-analysis, is there data on the cohort of infants who receive benefit from MgSO₄ exposure?

I cannot see the Koning et al 2018 (Int J Dev Neurosci) study in preterm rodents mentioned. How does this data relate to your results; ie preterm administration, and term-equivalent brain collection?

Have longer term preclinical studies to examine the effects of MgSO₄ been undertaken?

The piglet studies are mentioned in the Discussion but not the Introduction, which support a lack of neuroprotection with MgSO₄. Are the piglet studies representative of the preterm or term-equivalent brain?

The sample size for groups presented in the Results should be more clearly stated; Page 6 paragraph 3 states n=18 total fetuses; page paragraph 3 states HI+vehicle n=6, HI+MgSO₄ n=7, or sham n=6. It would be useful to add a line into the results to redefine the groups and sample sizes, and whether all fetuses were used in all analysis. It appears in some graphs (eg GFAP area) that n=7 fetuses were included in the sham group. Please clarify.

Is the MgSO₄ administered directly to the fetus? The Khatib et al 2022 (Placenta) study indicates that some benefits are mediated via a placental response. Is it possible that lack of positive results in the current study is due to route of administration? Please justify fetal administration and limitations.

Are the variable results to date confounded by dosing? Where do your fetal plasma levels sit in terms of dose? Can you correlate plasma MgSO₄ with specific effects in the brain?

END OF COMMENTS

HUDSON
INSTITUTE OF MEDICAL RESEARCH

MONASH University

THE UNIVERSITY OF AUCKLAND
NEW ZEALAND

Contact details:

Robert Galinsky, PhD.
The Ritchie Centre, Hudson Institute of Medical
Research and Department of Obstetrics and
Gynaecology, Monash University
Melbourne, 3162
Australia
Tel (+61 3) 8572 2866
E-mail: robert.galinsky@hudson.org.au

Monday, February 20, 2023

To: The Editors, J Physiol

Re: MgSO₄ reduces tertiary gliosis but does not improve EEG recovery, or white or grey matter cell survival after asphyxia in preterm fetal sheep

Authors: Robert Galinsky, Simerdeep K. Dhillon, Sharmony B. Kelly, Guido Wassink, Joanne O. Davidson, Christopher A. Lear, Lotte G. van den Heuij, Laura Bennet, Alistair J. Gunn

Dear Dr Schultz,

Thank you for the opportunity to respond to the editors' and reviewers' comments on our submission.

We have revised our manuscript in response to the editors' and reviewers' comments, provided a point-by-point response below and have uploaded all relevant files using the online submission system.

Sincerely,
Robert Galinsky and Alistair Jan Gunn
On behalf of the authors

Reviewing Editor:

This study investigates the role of MgSO₄ on brain development. Interestingly, although MgSO₄ improves EEG outcomes there is still loss of neurons. But this is an important study in understanding the underlying mechanisms for the potential benefits of such treatment.

The key points section may be over the word limit.

This section is within the word limit, word count: 140/150 words

Page 4 - Should 'neuroprotective' be 'neuroprotection'?

Corrected, thank you.

Page 4 - It would be helpful to mention glutamate in this paragraph. It comes into the discussion but associating it with NMDA here would improve clarity. It was confusing to read 'anti-excitatory'. Could rephrasing to something like '.. Mg inhibits the excitatory NT glutamate by inhibiting NMDA'?"

We have rephrased this section to read: The most likely mechanism for neuroprotection with magnesium is through its physiological role as an endogenous inhibitor of N-methyl-D-aspartate (NMDA) receptors by excitatory amino acids such as glutamate.

Page 6 - Were there nineteen animals in the study (6, 7, 6)?

Indeed, there were 19 animals in the study, thank you for alerting us to the typographical error.

Page 8 - It is stated that studies occurred between 103-125. Does this mean that day one could have been anywhere between 103-125 with tissue collection 21d later (124-146)? Or does this mean that every study started at d 103 and ended at d 125? If the former, this is a very wide range and brain development would be very different at tissue collection. If the later, please clarify.

The latter is correct. We have clarified this section accordingly, thank you.

Page 14 - JP does not have supplementary data.

We have now incorporated the supplementary figure and tables into the MS file as Figure 1 and Tables 1 and 2. Thank you

Page 17 - Please remove 'strongly' from before infer.

Removed, as requested.

It would be helpful to use line numbers.

Line numbers have been incorporated into the revised version of the MS.

Please include the sample size in the figure legends.

Figure legends now include a sample size. Thank you.

Senior Editor:

Comments to the Author:

Please state actual p values throughout including figures and include in figure legends the sample size(s) and statistical test(s) used in the figure.

Specific P values are now reported in the results and figures. The sample sizes and statistical tests used are now reported in the revised figure legends.

Referee #1:

Specific Comments

1. Page 4, line 20: 'Neuroprotection' not 'neuroprotective'.

Corrected. Thank you.

2. Page 7, line 12. Presumably, leads and catheters were exteriorized.

That is correct, we stated this in the Methods (page 7, paragraph 2)

3. Page 17, line 18. Where is the data to support the conclusion that MgSO₄ reduced total seizures in this study. This reviewer searched for the data and was unable to locate it. Please provide the data to support this conclusion in the current study.

We have clarified this section of the Discussion by stating we have previously shown that MgSO₄ reduces electrographic seizures using the same experimental paradigm and infusion protocol as the present study (page 18, paragraph 2). We apologise for any confusion this may have caused.

4. Did they not experience any fetal loss during the studies?

We have included a statement in the Methods to clarify that in this experimental protocol we experience a fetal loss rate of 20 % that did not vary between the experimental groups. In cases of fetal loss, the subject was excluded from study (page 9, paragraph 1).

Referee #2:

I found myself reading the Introduction a couple of times in an attempt for clarity on the rationale for the study. The way it is currently written, the study appears to be incremental given that you have previously demonstrated a lack of protection in preterm sheep with MgSO₄ in brains collected at 3 days. However I think this current study has merit to fill a knowledge gap on the effects of preterm exposure to MgSO₄ at term-equivalent brain age. We wholeheartedly agree with the reviewer and have revised the Introduction to clarify this important point (page 5, paragraph 2 and page 6, paragraph 1). Thank you.

In the Doyle 2009 meta-analysis, is there data on the cohort of infants who receive benefit from MgSO₄ exposure?

We have included the relative risk and confidence interval data that was reported in the meta-analysis for the cohort of infants who received benefit from MgSO₄ exposure in the revised Introduction (page 4, paragraph 1).

I cannot see the Koning et al 2018 (Int J Dev Neurosci) study in preterm rodents mentioned. How does this data relate to your results; ie preterm administration, and term-equivalent brain collection?

We thank the reviewer for pointing out the excellent paper by Koning et al. (Koning *et al.*, 2018) which showed a reduction in MBP loss with MgSO₄ treatment 7 days after hypoxia ischaemia. Although the serum Mg²⁺ concentration achieved in the study by Koning et al. (2.7-4.1 mmol/L) is much higher than the levels achieved in this study and in human cohort studies (1.89 mmol/L (page 13, paragraph 2)), the histological outcome showing an improvement in MBP+ staining with MgSO₄ is broadly consistent with our study. We have included this important study in the revised Discussion (page 23, paragraph 3).

Have longer term preclinical studies to examine the effects of MgSO₄ been undertaken?

To the best of our knowledge, this is the first study in a large animal translational model of preterm hypoxia ischaemia to evaluate the impact of MgSO₄ on EEG maturation, tertiary gliosis, and white and grey matter cell survival after 21-days recovery to term equivalent age. Similarly, in neonatal rats exposed to hypoxia ischaemia on postnatal day 4 (Daher *et al.*, 2018), after 40 days recovery postnatal MgSO₄ treatment was associated with reduced thalamic and hippocampal tissue loss in males only. However, the MgSO₄-induced reduction in tissue loss was not associated with significant improvements in motor function or cognition (Daher *et al.*, 2018). We have included this consideration in the revised Discussion (page 24, paragraph 2).

The piglet studies are mentioned in the Discussion but not the Introduction, which support a lack of neuroprotection with MgSO₄. Are the piglet studies representative of the preterm or term-equivalent brain?

The piglet studies are representative of the term-equivalent brain, we have clarified this in the revised Discussion (page 19, paragraph 2). Thank you.

The sample size for groups presented in the Results should be more clearly stated; Page 6 paragraph 3 states n=18 total fetuses; page paragraph 3 states HI+vehicle n=6, HI+MGSO₄ n=7, or sham n=6. It would be useful to add a line into the results to redefine the groups and sample sizes, and whether all fetuses were used in all analysis. It appears in some graphs (eg GFAP area) that n=7 fetuses were included in the sham group. Please clarify.

We apologise and thank the reviewer for alerting us to the typographical error. We have revised the methods, results and figure legends to clearly state the study comprised 19 subjects, with n=6 sham control, n=6 vehicle+occlusion and n=7 MgSO₄+occlusion.

Is the MgSO₄ administered directly to the fetus? The Khatib et al 2022 (Placenta) study indicates that some benefits are mediated via a placental response. Is it possible that lack of positive results in the current study is due to route of administration? Please justify fetal administration and limitations.

The reviewer raises an interesting and important point. We administered MgSO₄ directly to the fetus since Mg²⁺ transfer across the sheep placenta is limited (Akoury *et al.*, 1997). It is reasonable to expect that direct infusion to the fetus also exposes the placenta to MgSO₄, although this was not directly assessed in our study. We have incorporated this point into the revised Discussion (page 20, paragraph 2).

Indeed, it is possible that MgSO₄ promotes neuroprotection by modulating inflammation within the placenta, circulation or centrally. However, it is important to note that the study by Khatib et al. 2022 did not correlate placental inflammation with direct markers of brain inflammation or injury. Furthermore, two subgroup analyses focusing on infants exposed to clinical chorioamnionitis from a large randomised controlled trial (Rouse *et al.*, 2008) showed MgSO₄ was not associated with improvements in neurodevelopment at 2 years of age or reduced rates of intraventricular haemorrhage or periventricular leukomalacia (Kamyar *et al.*, 2016; Edwards *et al.*, 2018). Although, both data sets were relatively small, they suggest MgSO₄ did not modulate placental inflammation. Nevertheless, our data support the potential for MgSO₄ to promote central anti-inflammatory effects which were linked to a modest improvement in myelin density. We have included consideration of this important point in the revised Discussion (page 21, paragraph 2)

Are the variable results to date confounded by dosing? Where do your fetal plasma levels sit in terms of dose? Can you correlate plasma MgSO₄ with specific effects in the brain?

This is an interesting point, although it is something that our data set is unable to verify due to highly consistent fetal serum Mg²⁺ levels between subjects that were similar to cord blood levels in preterm infants after maternal MgSO₄ administration (page 23, paragraph 3). The dosing regime in this study produced fetal plasma levels of 1.89 ± 0.08 mmol/L in the MgSO₄+occlusion group compared to 0.88 ± 0.07 mmol/L in the vehicle+occlusion group (page 13, paragraph 2). Nevertheless, this is an important point to consider with respect to human cohort studies where the variable duration of MgSO₄ infusion relative to the timing of delivery could result in varying circulating magnesium levels in the fetus/neonate. We have included this point in the revised Discussion (page 21, paragraph 2).

Akoury HA, White SE, Homan JH, Cheung VY, Richardson BS & Bocking AD. (1997). Failure of magnesium sulfate infusion to inhibit uterine activity in pregnant sheep. *Am J Obstet Gynecol* **177**, 185-189.

Daher I, Le Dieu-Lugon B, Lecointre M, Dupré N, Voisin C, Leroux P, Dourmap N, Gonzalez BJ, Marret S, Leroux-Nicollet I & Cleren C. (2018). Time- and sex-dependent efficacy of magnesium sulfate to prevent behavioral impairments and cerebral damage in a mouse model of cerebral palsy. *Neurobiol Dis* **120**, 151-164.

Edwards JM, Edwards LE, Swamy GK & Grotegut CA. (2018). Magnesium sulfate for neuroprotection in the setting of chorioamnionitis. *J Matern Fetal Neonatal Med* **31**, 1156-1160.

Kamyar M, Manuck TA, Stoddard GJ, Varner MW & Clark E. (2016). Magnesium sulfate, chorioamnionitis, and neurodevelopment after preterm birth. *BJOG* **123**, 1161-1166.

Koning G, Lyngfelt E, Svedin P, Leverin AL, Jinnai M, Gressens P, Thornton C, Wang X, Mallard C & Hagberg H. (2018). Magnesium sulphate induces preconditioning in preterm rodent models of cerebral hypoxia-ischemia. *Int J Dev Neurosci* **70**, 56-66.

Rouse DJ, Hirtz DG, Thom E, Varner MW, Spong CY, Mercer BM, Iams JD, Wapner RJ, Sorokin Y, Alexander JM, Harper M, Thorp JM, Jr., Ramin SM, Malone FD, Carpenter M, Miodovnik M, Moawad A, O'Sullivan MJ, Peaceman AM, Hankins GD, Langer O, Caritis SN & Roberts JM. (2008). A randomized, controlled trial of magnesium sulfate for the prevention of cerebral palsy. *N Engl J Med* **359**, 895-905.

Dear Dr Galinsky,

Re: JP-RP-2023-284381R1 "MgSO₄ reduces tertiary gliosis but does not improve EEG recovery, or white or grey matter cell survival after asphyxia in preterm fetal sheep" by Robert Galinsky, Simerdeep Kaur Dhillon, Sharmony B Kelly, Guido Wassink, Joanne Davidson, Christopher Arthur Lear, Lotte G. van den Heuij, Laura Bennet, and Alistair J. Gunn

Thank you for submitting your revised Research Article to The Journal of Physiology. It has been assessed by the original Reviewing Editor and Referees and has been well received. Some final revisions have been requested.

Please address all the points raised and incorporate all requested revisions or explain in your Response to Referees why a change has not been made. We hope you will find the comments helpful and that you will be able to return your revised manuscript within 2 weeks. If you require longer than this, please contact journal staff: jp@physoc.org.

REVISION CHECKLIST:

We look forward to receiving your revised submission.

Yours sincerely,

Harold D Schultz
Senior Editor
The Journal of Physiology
<https://jp.msubmit.net>
<http://jp.physoc.org>
The Physiological Society
Hodgkin Huxley House
30 Farringdon Lane
London, EC1R 3AW
UK
<http://www.physoc.org>
<http://journals.physoc.org>

REQUIRED ITEMS FOR REVISION

The Journal of Physiology funds authors of provisionally accepted papers to use the premium BioRender site to create high resolution schematic figures. Follow this link and enter your details and the manuscript number to create and download figures. Upload these as the figure files for your revised submission. If you choose not to take up this offer we require figures to be of similar quality and resolution. If you are opting out of this service to authors, state this in the Comments section on the Detailed Information page of the submission form. The link provided should only be used for the purposes of this submission. Authors will be charged for figures created on this premium BioRender account if they are not related to this manuscript submission.

-Papers must comply with the Statistics Policy https://jp.msubmit.net/cgi-bin/main.plex?form_type=display_requirements#statistics

In summary:

-If $n \leq 30$, all data points must be plotted in the figure in a way that reveals their range and distribution. A bar graph with data points overlaid, a box and whisker plot or a violin plot (preferably with data points included) are acceptable formats.

-If $n > 30$, then the entire raw dataset must be made available either as supporting information, or hosted on a not-for-profit repository e.g. FigShare, with access details provided in the manuscript.

- n clearly defined (e.g. x cells from y slices in z animals) in the Methods. Authors should be mindful of pseudoreplication.

-All relevant n values must be clearly stated in the main text, figures and tables, and the Statistical Summary Document (required upon revision)

-The most appropriate summary statistic (e.g. mean or median and standard deviation) must be used. Standard Error of the Mean (SEM) alone is not permitted.

-Exact p values must be stated. Authors must not use 'greater than' or 'less than'. Exact p values must be stated to three significant figures even when 'no statistical significance' is claimed.

-Statistics Summary Document completed appropriately upon revision

EDITOR COMMENTS

Reviewing Editor:

Thank you for revising the paper.

Senior Editor:

If the statistical summary document has errors please describe what is incorrect: The statistical comparisons shown in the tables must be included in the Statistical Summary document to show the actual p values.

Comments to the Author:

The statistical comparisons shown in the tables must be included in the Statistical Summary document to show the actual p values.

REFEREE COMMENTS

Referee #1:

The authors have addressed all of the issues raised by the previous review. Consequently, the manuscript has been improved. This reviewer has no additional comments.

Referee #2:

Thank you for the opportunity to read this manuscript again. Authors have done very good job addressing queries and altering the manuscript with reviewer comments in mind.

END OF COMMENTS

1st Confidential Review

21-Feb-2023

HUDSON
INSTITUTE OF MEDICAL RESEARCH

 **MONASH** University

THE UNIVERSITY OF AUCKLAND
NEW ZEALAND

Contact details:

Robert Galinsky, PhD.
The Ritchie Centre, Hudson Institute of Medical
Research and Department of Obstetrics and
Gynaecology, Monash University
Melbourne, 3162
Australia
Tel (+61 3) 8572 2866
E-mail: robert.galinsky@hudson.org.au

Thursday, March 09, 2023

To: The Editors, J Physiol

Re: MgSO₄ reduces tertiary gliosis but does not improve EEG recovery, or white or grey matter cell survival after asphyxia in preterm fetal sheep

Authors: Robert Galinsky, Simerdeep K. Dhillon, Sharmony B. Kelly, Guido Wassink, Joanne O. Davidson, Christopher A. Lear, Lotte G. van den Heuij, Laura Bennet, Alistair J. Gunn

Dear Dr Schultz,

As requested, we have now included the statistical comparisons included in the tables in the revised statistical summary document. Actual p values to 4 decimal places are reported.

Sincerely,
Robert Galinsky and Alistair Jan Gunn
On behalf of the authors

Dear Dr Galinsky,

Re: JP-RP-2023-284381R2 "MgSO₄ reduces tertiary gliosis but does not improve EEG recovery, or white or grey matter cell survival after asphyxia in preterm fetal sheep" by Robert Galinsky, Simerdeep Kaur Dhillon, Sharmony B Kelly, Guido Wassink, Joanne Davidson, Christopher Arthur Lear, Lotte G. van den Heuij, Laura Bennet, and Alistair J. Gunn

We are pleased to tell you that your paper has been accepted for publication in The Journal of Physiology.

Authors should note that it is too late at this point to offer corrections prior to proofing. The accepted version will be published online, ahead of the copy edited and typeset version being made available. Major corrections at proof stage, such as changes to figures, will be referred to the Editors for approval before they can be incorporated. Only minor changes, such as to style and consistency, should be made at proof stage. Changes that need to be made after proof stage will usually require a formal correction notice.

Yours sincerely,

Harold D Schultz
Senior Editor
The Journal of Physiology
<https://jp.msubmit.net>
<http://jp.physoc.org>
The Physiological Society
Hodgkin Huxley House
30 Farringdon Lane
London, EC1R 3AW
UK
<http://www.physoc.org>
<http://journals.physoc.org>

P.S. - You can help your research get the attention it deserves! Check out Wiley's free Promotion Guide for best-practice recommendations for promoting your work at www.wileyauthors.com/eeo/guide. You can learn more about Wiley Editing Services which offers professional video, design, and writing services to create shareable video abstracts, infographics, conference posters, lay summaries, and research news stories for your research at www.wileyauthors.com/eeo/promotion.

IMPORTANT NOTICE ABOUT OPEN ACCESS: To assist authors whose funding agencies mandate public access to published research findings sooner than 12 months after publication, The Journal of Physiology allows authors to pay an Open Access (OA) fee to have their papers made freely available immediately on publication.

You can check if your funder or institution has a Wiley Open Access Account here: <https://authorservices.wiley.com/author-resources/Journal-Authors/licensing-and-open-access/open-access/author-compliance-tool.html>.

EDITOR COMMENTS

Thank you for completion of the statistical document. We are pleased to accept your excellent study for publication in the Journal of Physiology.